# A Sodium Oxalate-Rich Diet Induces Chronic Kidney Disease and Cardiac Dysfunction in Rats

**DOI:** 10.3390/ijms22179244

**Published:** 2021-08-26

**Authors:** Thayane Crestani, Renato O. Crajoinas, Leonardo Jensen, Leno L. Dima, Perrine Burdeyron, Thierry Hauet, Sebastien Giraud, Clara Steichen

**Affiliations:** 1Laboratory of Genetics and Molecular Cardiology, Heart Institute, University of São Paulo Medical School, São Paulo 05403-900, SP, Brazil; thayc@hotmail.com (T.C.); renatocrajoinas@mail.com (R.O.C.); jensenleonardo@gmail.com (L.J.); lenodima@gmail.com (L.L.D.); 2INSERM U1082 (IRTOMIT), F-86000 Poitiers, France; perrine.burdeyron@univ-poitiers.fr (P.B.); thierry.hauet@univ-poitiers.fr (T.H.); giraudseb@yahoo.fr (S.G.); 3Faculté de Médecine et Pharmacie, Université de Poitiers, F-86000 Poitiers, France; 4Service de Biochimie, CHU Poitiers, F-86000 Poitiers, France

**Keywords:** chronic kidney disease, rat model, oxalate, cardiac function

## Abstract

Chronic kidney disease (CKD) is a worldwide public health issue affecting 14% of the general population. However, research focusing on CKD mechanisms/treatment is limited because of a lack of animal models recapitulating the disease physiopathology, including its complications. We analyzed the effects of a three-week diet rich in sodium oxalate (OXA diet) on rats and showed that, compared to controls, rats developed a stable CKD with a 60% reduction in glomerular filtration rate, elevated blood urea levels and proteinuria. Histological analyses revealed massive cortical disorganization, tubular atrophy and fibrosis. Males and females were sensitive to the OXA diet, but decreasing the diet period to one week led to GFR significance but not stable diminution. Rats treated with the OXA diet also displayed classical CKD complications such as elevated blood pressure and reduced hematocrit. Functional cardiac analyses revealed that the OXA diet triggered significant cardiac dysfunction. Altogether, our results showed the feasibility of using a convenient and non-invasive strategy to induce CKD and its classical systemic complications in rats. This model, which avoids kidney mass loss or acute toxicity, has strong potential for research into CKD mechanisms and novel therapies, which could protect and postpone the use of dialysis or transplantation.

## 1. Introduction

Chronic kidney disease (CKD) is a major public health issue, with an overall prevalence in the general population of approximately 14% [1]. CKD is characterized by an impairment of kidney filtration function caused by a progressive loss of nephrons, which are replaced by interstitial fibrosis. Other than transplantations that are restricted for end-stage CKD and impaired by worldwide organ shortages, therapeutic strategies are limited to dialysis, which is efficient but significantly affects a patient’s social and psychological life, as well as being extremely expensive for the social care system. In this context, the availability of accurate models of CKD is mandatory to better understand the disease, control its progression and potentially postpone its complications and the necessity of dialysis and transplantation.

Hyperoxaluria is one of the primary risk factors for the occurrence of oxalate crystals. Hyperoxaluria may be influenced by specific diets [2], such as vegetarianism [3,4], which is becoming common, or by gut microbiome composition [3,4,5]. Currently, it is widely accepted that these crystals induce kidney stones and systemic disorders, including chronic kidney diseases and renal failure, metabolic disorder and cardiovascular diseases [4,6,7,8,9,10].

Therefore, translational research in the area of CKD requires adequate experimental models, and the development of alternative therapeutic strategies is of crucial importance; however, these strategies are conditioned by access to animal models that closely recapitulate the physiopathology of the disease. Indeed, CKD animal models are numerous, but the quest for a model that most accurately recapitulate human conditions is still ongoing [11]. Other than CKD induced by toxic agents, sub-total nephrectomies such as 5/6 are established methods to model progressive renal failure, and these mimic the consequences of nephron reduction but not necessarily renal disease. Moreover, CKD in patients is also associated with its complications such as elevated blood pressure and cardiac issues [9]. It has been shown that a diet rich in sodium oxalate induces stable CKD in mice, as well as related hypertension and cardiac fibrosis [12]. However, the use of mice can be limiting for specific applications such as testing surgery procedures or applications where a large amount of kidney material is needed. Importantly, the observed evolution of human health demographics, with an increasing number of obese, hypertensive and diabetic individuals, is a major issue; thus, a rat model allows us to consider rat species that naturally develop these pathologies (such as: Zucker rat, spontaneously hypertensive rat, Brattleboro rat and biobreeding diabetes-prone rat) and may also be confronted with oxalate-rich diets. Indeed, both diabetes mellitus and obesity are associated with higher urinary oxalate excretion, both of which are mediators of severe kidney disease [13]. This study investigates the impact of a high sodium oxalate diet on rats by assessing both renal structure and function and cardiovascular complications.

## 2. Results

### 2.1. High Oxalate Diet Significantly Impaired Rats’ Biological Parameters and Kidney Function

Our results showed that food consumption was slightly reduced during the 3 weeks of an OXA diet, and was linked with a slower gain of weight (Figure 1a,b); however, water consumption and urinary flux increased with the OXA diet (Figure 1c,d).

The impact of the OXA diet on kidney function was assessed further. First, animal glomerular filtration rate (GFR) was reduced from 60% after 3 weeks of the OXA diet compared to the control diet (CTR diet) (Figure 2a,b, *p* = 0.005). This decrease was already visible after 1 week of the OXA diet, and it stayed stable for 2 additional weeks following oxalate retrieval on day 21. This reduced GFR was linked with elevated plasmatic creatinine and urea levels (Figure 2c,d, *p*-values = 0.0041 and 0.0045, respectively) in the OXA group compared to the CTR one. In contrast, total protein/creatinine ratio was not significantly modified (Figure 2e), but rats displayed proteinuria, notably albuminuria, after 3 weeks of the OXA diet (Figure 2f), with a peak on day 21.

### 2.2. High Oxalate Diet Induced Tubular Atrophy and Fibrosis though Oxalate Crystals’ Deposition in the Tubular Lumen

At the end of the 3-week diet (D21), alterations in kidney morphology induced by the OXA diet were visible macroscopically with the presence of “white spots” all over the kidney parenchyma. At a higher magnification, oxalate crystals were visible under polarized light (Figure 3a). Histological analyses using periodic acid shift staining showed a large disorganization of the cortex with several atrophied tubules (black arrows) and presence of fibrosis with collagen deposition (black arrow) (Figure 3b). We confirmed the presence of oxalate crystals within the tubular lumen of atrophied tubule (Figure 3c).

### 2.3. Modulation of High Oxalate Diet

We then evaluated whether CKD induction by the OXA diet was altered by modulating diet duration, sodium oxalate concentration and rat gender. First, one week of an oxalate diet was sufficient to impair kidney function, with rats displaying lower GFR and higher blood urea nitrogen (BUN) levels; however, this impairment was not substantial after oxalate retrieval. During this one-week regimen, sodium oxalate used at three different concentrations (25, 50 and 100 µmol/g) had a similar effect on kidney parameters (Appendix A, Figure A1). Importantly, both females and males seem to be equally sensitive to a 3-week high oxalate diet (at 50 µmol/g) (Appendix A, Figure A2).

### 2.4. Effect of Oxalate Diet on Systemic Complications and Cardiovascular System

As well as impairments of kidney function, we next analyzed the impact of CKD on systemic parameters after the 3-week diet (D21). First, the OXA diet induced a significant decrease in hematocrit and hemoglobin concentrations (*p* < 0.01), as well as blood glutamate concentration (*p* < 0.05), compared to control rats (Figure 4). However, blood pH, potassium, glucose concentration, as well as other biological parameters we analyzed were not significantly modified by the OXA diet (Table 1).

In addition, at the end of the 3-week diet (D21), caudal artery measurements of blood pressure showed that the OXA diet induced arterial hypertension (*p*-value = 0.0003) (Figure 5a). Moreover, cardiac function was monitored by echocardiography at the end of the treatment (D21). Heart rate was not modified by 3 weeks of the OXA diet (Table 2), but we observed a significant increase in intraventricular septum (IVS) thickness during diastole (IVS;d) (*p*-value = 0.0325), as well as a trend towards an increased left ventricular (LV) mass in the OXA group (*p*-value = 0.0594) (Figure 5b,c). Moreover, the OXA diet resulted in a modified (E/e’) ratio (*p*-value = 0.0366) (Figure 5d and Table 2), suggesting abnormal diastolic cardiac function.

## 3. Discussion

Our results showed that the use of a noninvasive procedure based on a sodium oxalate-rich diet triggered CKD in rats, as well as related systemic complications. Indeed, using 50 µmol/g of oxalate for 3 weeks induced a significant GFR decrease of 60%, corresponding to stage 3 CKD in human patients. Interestingly, this GFR reduction, as well as BUN levels, remained stable despite removal of sodium oxalate from the diet. Modulating the sodium oxalate dose (from 25 to 100 µmol/g) did not influence the severity of the condition. However, reducing the diet duration to 1 week was not sufficient to sustain the GFR decrease, showing that kidney dysfunction is progressive and irreversible over the 3-week regimen. In addition, we showed that 3 weeks of the OXA diet also negatively impacted systemic parameters and heart function with elevated blood pressure, and also decreased hematocrit and cardiac diastolic dysfunction. These results are important, since this model can be used to study both mechanisms and treatments for CKD and associated complications, closely mimicking the progression of human pathology.

Oxalate nephropathy is a rare condition affecting patients, and it is caused by either inherited enzymatic deficiencies (primary hyperoxaluria) or exogenous sources, such as dietary consumption of products rich in oxalate, as well as via other substances that may metabolize into oxalate within the body, such as vitamin C (secondary hyperoxaluria) [14,15]. Stricto sensu, our model mimicked secondary hyperoxaluria, since sodium oxalate was facilitated through the rats’ diets. However, phenotypic consequences of this OXA diet model, including progressivity and sustainability of the kidney impairment, are those of CKD. Removal of calcium from the diet increases the availability of the oxalate and enhances its potential to be crystallized [16], but we did not test the effect of high sodium oxalate concentrations in combination with a classical diet. However, we showed that our so-called control diet (low-calcium diet) used over a 3 to 5 week period already had a slight impact on kidney function, as shown in Figure 2, and it may be interesting to further investigate the long-term effects of a low-calcium diet on rat kidney function in a future study. Interestingly, acid/base balance was not significantly modified by the OXA diet in our model. In chronic kidney diseases, with declining kidney function, acid retention and metabolic acidosis occur, but the extent of acid retention depends on the degree of kidney impairment (Nagami and Hamm, 2017; Siener, 2018). We observed a relative functional decline, and this can support the limited alteration in the acid/base balance. Moreover, acid retention can occur even when the serum bicarbonate level is apparently normal.

In vitro, it has been shown that oxalate at supraphysiologic concentrations induces epithelial cells necrosis [17] and negatively impacts growth and survival of renal epithelial and endothelial cells, as well as fibroblasts with interstitial calcification, notably within the interstitial space [18,19]. This is probably due to a cellular mechanism that links oxalate and necroptosis induced by mitochondria permeability transition [20]. We did not observe any crystals in the interstitium, but we did observe them within the tubular lumen of atrophied tubules. When used on mice, the OXA diet also induced hyperphosphatemia, hyperkalemia and metabolic acidosis, in addition to kidney impairment [12]. Interestingly, these parameters were not altered in our model, suggesting a partial inter-species difference. Gender contributes to differences in incidence and progression of CKD, and some rat models, such as adenine-induced CKD, displayed differences, with males being more severely affected than females [21], at least partially due to estrogen/estrogen receptors [22,23]. After a dose–response study, which allowed us to choose an oxalate concentration of 50 µmol/g (Figure A1), all of our experiments were performed on males, excluding one comparative experiment that may have shown that both male and females were sensitive to the OXA diet; this is despite there not being a significant trend for more severe phenotypes in females (Figure A2), suggesting that gender does not influence oxalate nephrotoxicity, which is a major advantage of our protocol.

We hypothesize that this rat model is specifically suitable for the testing of drugs or cell therapy strategies. Indeed, total kidney volume is maintained (no nephrectomy) but it also displays atrophied tubules, leaving a space for some candidate cells to graft and eventually repopulate part of the nephrons. In addition, internal capacity of the kidney to repair may also be modulated using drugs or stem cell-derived products such as microvesicules. It is important to highlight that, in cases local therapeutic intervention, the contralateral kidney is available and can serve as an internal control (without cell or drug injections for example). Finally, using rats also allows for the disposal of a sufficient quantity of kidney or tissue in order to perform several downstream analyses per animal, compared to mouse models. Moreover, this model may be applied to rat species that display comorbidities, taking into account the physiopathology complexity that often relies on a combination of factors. Finally, our proposed model uses a common and suitable animal model, the rat, to induce stable CKD within a limiting time using a simple and cheap diet intervention. This model works for both male and female rats, does not require surgery or a reduction in renal mass and significantly impacts GFR and typical CKD complications; therefore, this model is important in relation to CKD research and cardiovascular disease in general. Since this model induces chronic kidney disease, it is complementary to existing models that use sodium oxalate in rats (intravenous injection) to induce acute kidney injury, with reported glomerular and tubulo-interstitial damages after 24 h of treatment [24].

Furthermore, the OXA diet induced persistent arterial hypertension, resulting in a higher intraventricular septum thickness during diastole, which corresponded to LV hypertrophy without alterations in the cavity diameters. This fact corroborates the trend towards increased LV mass. Diastolic flow analysis revealed that animals in the OXA diet group also presented diastolic dysfunction due to a higher ratio between mitral flow and tissue flow (E/e’). This leads us to believe that the OXA diet is linked to an early stage of diastolic dysfunction with a phase of compensated cardiac hypertrophy [25,26].

Please note that the unlimited access to food may lead to differences in food intake between animals, which may be associated with phenotypic differences; some animals did not develop frank cardiac hypertrophy as expected. It is possible that, by controlling access to an oxalate diet or using oral administration of OXA by gavage, the parameters that presented a statistical trend could in fact be significantly different. In addition, extending OXA administration may lead to more pronounced cardiac damage.

In conclusion, we reported on an efficient, highly convenient and non-invasive model of CKD in rats based on a diet rich in sodium oxalate that resulted in (i) a significant and stable reduction in renal function; (ii) a CKD within a relatively short time to limit the burden for animals and reduce housing costs; (iii) classical systemic complications and cardiac dysfunction; (iv) a model without the need for surgery (nephrectomy) or anesthetic/analgesic drugs, which could induce confounding effects; and (v) a model without a sex/gender impact.

This model has appropriate applicability for the evaluation of novel therapeutic strategies, which may include (i) therapy with the administration of an extract from the *Grona styracifolia* plant [27], or microorganisms such as *O. formigenes*, which utilizes oxalate as its predominant source of energy [28,29]; (ii) therapy with inhibitors such as the long pentraxin 3 (PTX3) [30], anti-necroptosis [20] or blockade of TNF (tumor necrosis factor) receptor (TNFR) [31]; (iii) deletion of inflammasome proteins [32,33]; (iv) or cell-based therapies with the advantage that both kidneys are present and affected by the disease and one can be used as a control for local interventions. The strategies may be efficient enough to limit CKD progression and potentially postpone its complications, as well as the necessity of dialysis or transplantation.

## 4. Materials and Methods

### 4.1. Animal Studies

Animal studies were carried out in accordance with the ethical principles in animal research of the Brazilian College of Animal Experimentation, and were approved by the Institutional Animal Care and Use Committee. Male and female Wistar 6–7 week-old rats were obtained from animal care of Heart Institute and housed in groups of 4 in standard housing conditions with unlimited access to food and water. Kidney function was assessed at the initiation of the diet and then weekly, placing each rat in independent metabolic cages where we quantified 24 h urine and feces production, as well as 24 h food and water consumption. Rats were weighted weekly when entering metabolic cages and submitted to retro-orbital sampling of blood. To ensure that stress induced by metabolic cages had no impact, rats were placed in metabolic cages for a 24 h adaptation period before the first measurement time. The oxalate diet was prepared by mixing 50 µmol/g of sodium oxalate to a low-calcium diet (Rhoster, Araçoiaba da Serra, Brazil), with the latter being named as the control diet. Removal of calcium from the diet increased the amount of soluble oxalate and therefore the impact of the diet [16].

Experimental design is presented in detail in Figure 6. The design enabled the weekly monitoring of rat health and kidney function and the analysis of the impact of a regimen composed of a 3-week high-oxalate diet (OXA diet), followed (or not) by a 2-week control diet (CTR diet) on rat biological parameters.

### 4.2. Assessment of Renal Function

Urine was collected (U in Figure 6) in metabolic cages and stored aliquoted at −20 °C. Blood arterial samples were harvested (B in Figure 6) by retro-orbital sampling and stored aliquoted at −80 °C. Creatinine, urea and protein concentrations in urine were quantified on fresh samples using commercially available kits (Labtest, Lagoa Santa, Brazil) following the manufacturer’s instructions. GFR was calculated using the following formula: GFR = ((creatinine concentration in urine) × urine flow rate)/(creatinine concentration in plasma) and then normalized per 100 g of body weight. Arterial blood was collected for measurements of pH, blood gas, electrolyte, oximetry and metabolite parameters by a blood gas analyzer (ABL800 Flex, Neuilly-Plaisance, Radiometer). Urinary creatinine concentrations were measured using a kinetic method (Labtest). Volumes of urine containing 5 mg of creatinine were solubilized in Laemmli sample buffer and resolved using 10% SDS-PAGE gels. Following electrophoresis, gels containing proteins from urine samples were silver stained using the ProteoSilver Plus kit (Merck, Darmstadt, Germany) to detect urinary proteins.

### 4.3. Blood Pressure Measurement

Blood pressure were measured in conscious animals by a pressure transducer (model DT-100; Utah Medical Products, Midvale, UT, USA) and recorded using an interface and software for computer data acquisition (Power Lab 4/25; AD Instruments, Sydney, Australia), as described previously [34]. Twenty-four hours before measurements, rats were trained with the blood pressure device to adapt to the experimental procedures.

### 4.4. Echocardiographic Assessments

Cardiac function was monitored by high frequency ultrasound VEVO 2100 (Visual Sonics, Canada) at the end of the treatment. At D21 (E in Figure 6), rats in both groups were anesthetized with 2% of isoflurane and placed in the left lateral decubitus position to obtain cardiac images. Left ventricular mass was calculated using Penn equation [35]. Linear measures were obtained at parasternal short axis view of the left ventricle (LV) with M-mode. Doppler analyses and tissue Doppler were obtained by apical axis at four-chamber view. Measurements were performed as recommended by the American Society of Echocardiography, and were conducted by an experienced echocardiographist who was blinded for study groups.

### 4.5. Histological Analyses

Histological analysis was performed at the end of the treatment, at D21 (H in Figure 6). After rat euthanasias using isoflurane inhalation, kidney and heart were harvested, weighed, rinsed in saline solution and placed in paraformaldehyde (PFA) 4% overnight before inclusion in paraffin. Then, 2 µM sections were stained with eosin/hematoxylin, periodic acid shift and Masson trichrome.

### 4.6. Statistical Analyses

Results are presented as kinetics to emphasize the diet’s impact, but statistical analysis was performed using Student’s t-test to compare rats submitted to the OXA diet versus control diet, at each time point independently.

## Figures and Tables

**Figure 1 ijms-22-09244-f001:**
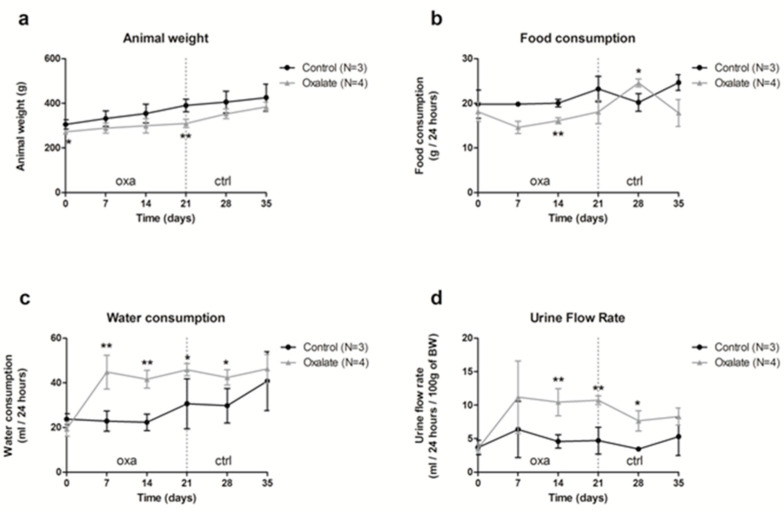
Impact of the 5-week regimen (3 weeks of OXA diet + 2 weeks of control diet) on metabolic parameters: (**a**) animal weight (grams); (**b**) food consumption (grams/24-h); (**c**) water consumption (mL/24-h); (**d**) urine flow rate (mL/24-h/100 g of body weight). Results are expressed as mean ± SD. * *p* ≤ 0.05; ** *p* ≤ 0.01, versus control. Control *n* = 3, oxalate *n* = 4.

**Figure 2 ijms-22-09244-f002:**
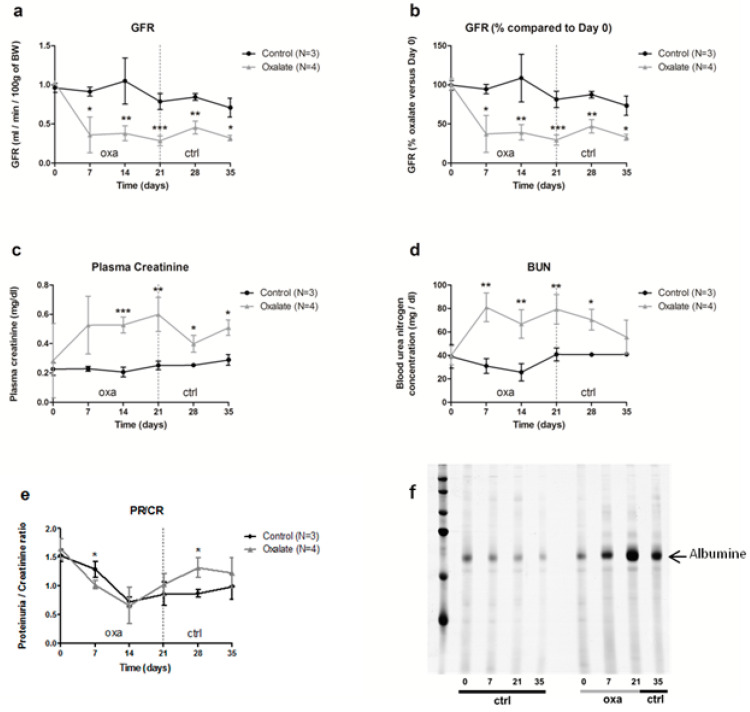
Impact of the 5-week regimen (3 weeks of OXA diet + 2 weeks of control diet) on kidney function: (**a**) glomerular filtration rate (mL/min/100 g of body weight); (**b**): percentage of GFR compared to day 0 of the diet (%); (**c**) plasma creatinine concentration (mg/dL); (**d**) blood urea nitrogen concentration (mg/dL); (**e**) ratio total protein/creatinine concentration (mg/24-h/100 g of body weight); (**f**) profile of urinary proteins excreted by ctrl and oxa rats. The 24 h urine samples were subjected to 10% SDS-PAGE. Following electrophoresis, the gels were silver stained using the ProteoSilver Plus Kit (Sigma-Aldrich). Results are expressed as mean ± SD. * *p* ≤ 0.05; ** *p* ≤ 0.01; *** *p* ≤ 0.001, versus control. Control *n* = 3, oxalate *n* = 4.

**Figure 3 ijms-22-09244-f003:**
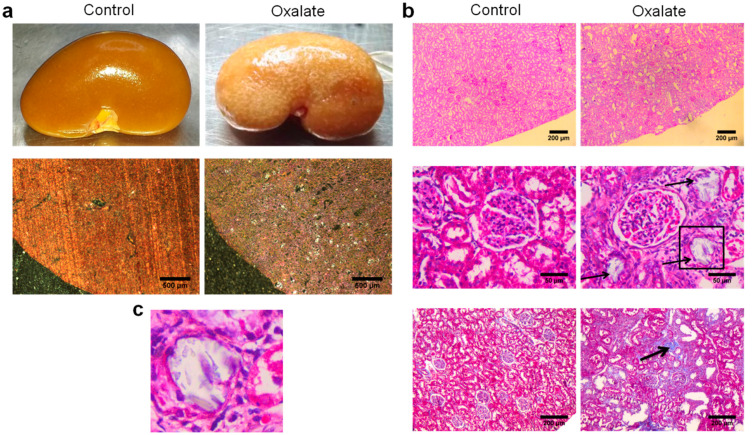
Representative morphological analyses of the kidneys of rats treated with OXA diet compared to controls: (**a**) morphology of the right kidney after its retrieval (upper line) and observation of kidney section under polarized light showing presence of crystals within the cortex; (**b**) periodic acid shift staining of sliced kidney showing massive cortical disorganization (upper line), presence of several atrophied tubules (middle line) and areas of fibrosis (lower line); (**c**) enlargement of the atrophied tubule highlighted with a black square in subfigure b showing the presence of crystals within the tubular lumen.

**Figure 4 ijms-22-09244-f004:**
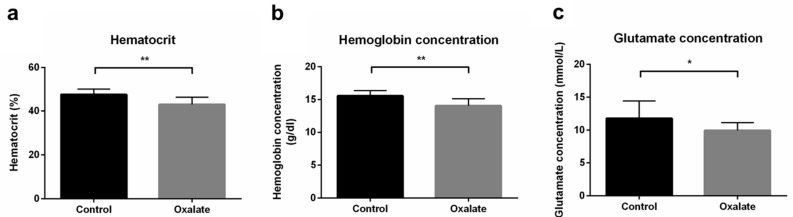
Impact of the 3-week OXA diet on systemic parameters: (**a**) hematocrit (%); (**b**) hemoglobin concentration (g/dL); (**c**) glutamate concentration (mmol/L). Results are expressed as mean ± SD. * *p* ≤ 0.05; ** *p* ≤ 0.01. Control *n* = 6, oxalate *n* = 18.

**Figure 5 ijms-22-09244-f005:**
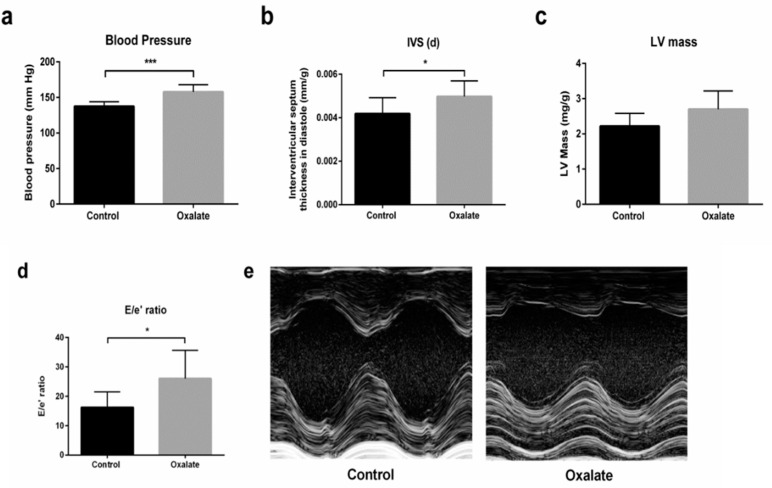
Impact of 3 weeks of an OXA diet on blood pressure and cardiac function: (**a**) blood pressure (mmHg); (**b**) IVS;d (mm/kg)—intraventricular diastolic septum thickness; (**c**): LV mass (mg/kg)—left ventricular mass; (**d**) E/e’ ratio; (**e**) representative images in both experimental groups at parasternal short axis view of the LV with M-mode. Results are expressed as mean ± SD. * *p* ≤ 0.05; *** *p* ≤ 0.001. Control *n* = 5, oxalate *n* = 27.

**Figure 6 ijms-22-09244-f006:**
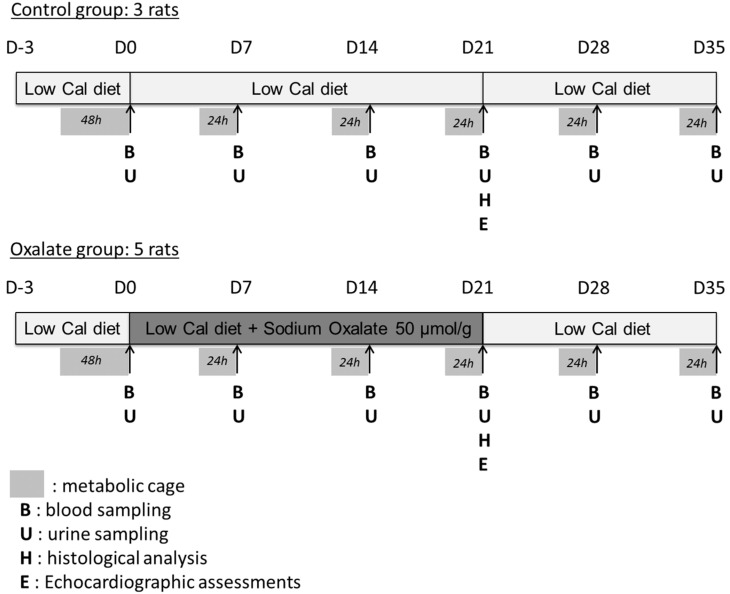
Experimental design of the 5-week regimen for both control and oxalate groups.

**Table 1 ijms-22-09244-t001:** Blood parameter analysis after 3 weeks of the OXA diet (results are expressed as mean ± SD. Control *n* = 6, oxalate *n* = 18).

	Unit	Control (*n* = 6)	Oxalate (*n* = 18)	*p* Value
**Oxygen Status**				
Oxygen partial pressure (*p*O_2_)	mmHg	56.82 ± 4.41	58.44 ± 2.12	0.717
Carbon dioxide partial pressure (*p*CO_2_)	mmHg	39.60 ± 1.01	34.29 ± 1.51	0.062
Oxygen tension at half saturationof blood (*p*50)	mmHg	33.43 ± 1.79	31.21 ± 0.94	0.260
Oxygen saturation (*s*O_2_)	%	79.53 ± 4.06	84.49 ± 1.41	0.152
Hematocrit	%	47.60 ± 1.30	43.07 ± 0.77	0.006 **
Hemoglobin	g/dL	15.53 ± 0.34	14.03 ± 0.26	0.006 **
Fraction of oxyhemoglobin (FO_2_Hb)	%	78.88 ± 3.90	83.26 ± 1.31	0.178
Fraction of deoxyhemoglobin (FHHb)	%	20.33 ± 4.05	15.38 ± 1.40	0.150
Oxygen content (ctO_2_)	Vol%	17.22 ± 0.98	16.58 ± 0.46	0.516
**Electrolytes concentration**				
Sodium (Na^+^)	mmol/L	137.80 ± 1.81	139.60 ± 1.32	0.502
Potassium (K^+^)	mmol/L	4.05 ± 0.19	3.80 ± 0.06	0.109
Ionized calcium (Ca^2+^)	mmol/L	1.16 ± 0.05	1.08 ± 0.03	0.194
Chloride (Cl^‒^)	mmol/L	119.80 ± 3.89	124.90 ± 3.70	0.343
**Metabolite concentrations**				
Glutamate	mmol/L	11.77 ± 1.09	9.92 ± 0.28	0.027 *
Lactate	mmol/L	2.18 ± 0.33	2.56 ± 0.18	0.330
Glucose	mg/dL	106.69 ± 39.79	105.34 ± 45.10	0.969
**Acid/base status**				
Standard base excess (cBase(Ecf))	mmol/L	−9.32 ± 1.04	−9.58 ± 0.72	0.852
Standard bicarbonate (cHCO_3_^‒^(*p*,st))	mmol/L	16.40 ± 0.829	16.70 ± 0.460	0.7498
Blood pH		7.245 ± 0.023	7.287 ± 0.009	0.0520.

*: *p* ≤ 0.05; **: *p* ≤ 0.01.

**Table 2 ijms-22-09244-t002:** Echocardiographic parameter analysis after 3 weeks of OXA diet (results are expressed as mean ± SD. Control *n* = 5, oxalate *n* = 27).

	Unit	Control (*n* = 5)	Oxalate (*n* = 27)	*p* Value
Heart Rate	BPM	366.20 ± 7.55	359.0 ± 7.746	0.7012
Left Ventricle Diameter (d)	mm/g	0.023 ± 0.0008	0.024 ± 0.0005	0.582
Left Ventricle Diameter (s)	mm/g	0.0127 ± 0.0008	0.014 ± 0.0006	0.345
Interventricular septum tickness (d)	mm/g	0.0042 ± 0.0003	0.005 ± 0.0001	0.032
Interventricular septum tickness (s)	mm/g	0.00797 ± 0.0008	0.0083 ± 0.0002	0.630
LV Mass	mg/g	2.22 ± 0.16	2.70 ± 0.102	0.0601
Volume (s)	µl	117.00 ± 7.84	133.8 ± 12.40	0.5643
Volume (d)	µl	389.4 ± 34.59	449.7 ± 20.05	0.2231
Ejection fraction	%	68.99 ± 3.592	70.43 ± 2.136	0.7810
Fractional shortening	%	19.25 ± 1.956	21.66 ± 0.924	0.2932
Stroke Volume	µl	272.4 ± 34.24	313.9 ± 15.66	0.2937
E-wave Velocity	mm/s	991.3 ± 53.99	1030 ± 26.20	0.5558
A-wave Velocity	mm/s	516.0 ± 42.95	547.5 ± 23.94	0.5966
Isovolumetric relaxation time	ms	23.11 ± 1.174	23.08 ± 0.605	0.9845
Deceleration Time	ms	28.22 ± 4.247	28.44 ± 1.544	0.9572
E’-wave Velocity	mm/s * BPM^−1^	0.18 ± 0.067	0.13 ± 0.010	0.0421 *
A’-wave Velocity	mm/s * BPM^−1^	0.13 ± 0.023	0.15 ± 0.009	0.5861
A’/E’ ratio	NA	0.88 ± 0.292	1.40 ± 0.148	0.1702
E’/A’ ratio	NA	1.60 ± 0.393	0.99 ± 0.111	0.0556
E/A ratio	NA	1.98 ± 0.208	0.19 ± 0.095	0.8772
E/E’ ratio	NA	16.29 ± 3.352	26.04 ± 1.851	0.0366 *

*: *p* ≤ 0.05.

## Data Availability

The data presented in this study are available on request from the corresponding author.

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
