# Peer review of "A Sodium Oxalate-Rich Diet Induces Chronic Kidney Disease and Cardiac Dysfunction in Rats"

_ijms, 2021, doi:10.3390/ijms22179244_

Round 1
Reviewer 1 Report
Crestani et. al. examined sodium oxalate (OXA)-rich diet on rats to demonstrate kidney damage accompanied with anemia, hypertension, and cardiac dysfunction. This can be a good animal model for CKD and the complications.
Although the manuscript was of interest, a couple of issues should be concerned.
- In Fig. 2f, authors showed that the 24-h urine samples were subjected to 10% SDS-PAGE. How did they apply the urine samples? The method is absent.
- In Fig. 3, the kidneys look similar in size. However, how was the kidney size or weight in the OXA diet model rats, compared to the control?
- In the OXA diet model, authors showed cardiac dysfunction, which was interesting. However, is the cardiac dysfunction resulted directly from OXA or mediated with decreased GFR? Did authors look at the heart tissue?
- In Fig A2, the number is too small to assess the sex difference. Authors concluded the major advantage of this model without sex difference in Line 195-198 and Line 233-234 solely based on the results, which is scientifically vulnerable.
- What was the serum concentration of phosphate and parathyroid hormone (PTH)? Secondary hyperparathyroidism is also the major complication of CKD.
Author Response
Please find the point by point answer in the word document.

Reviewer 2 Report
I find it really amazing that the methods section is located at the end of the manuscript, simply I believe it is not its natural location.
The fact that Oxa induced kidney disease without altering the acid-base balance is very surprising and should be better discussed.
When using latin citations (such as in row 173) it is mandatory to cite them in correct form. "stricto sensus" is wrong, the correct way is "stricto sensu" (both terms in ablative).
References seem "dated", I suggest the authors to add more recent papers.
Author Response
Please find a point by point response in the word document.

Round 2
Reviewer 1 Report
Authors appropriately answered to my questions with corrected manuscript.